# Evolution of Cherries (*Prunus* Subgenus *Cerasus*) Based on Chloroplast Genomes

**DOI:** 10.3390/ijms242115612

**Published:** 2023-10-26

**Authors:** Xin Shen, Wenjin Zong, Yingang Li, Xinhong Liu, Fei Zhuge, Qi Zhou, Shiliang Zhou, Dongyue Jiang

**Affiliations:** 1Institute of Tree Breeding, Zhejiang Academy of Forestry, 399 Liuhe Road, Hangzhou 310023, China; shenxinjdy@126.com (X.S.); zwj2166@163.com (W.Z.); hzliyg@126.com (Y.L.); lsliuxh@163.com (X.L.); zhugefei135@163.com (F.Z.); qizhou36@hotmail.com (Q.Z.); 2State Key Laboratory of Systematic & Evolutionary Botany, Institute of Botany, Chinese Academy of Sciences, Beijing 100093, China

**Keywords:** biogeography, chloroplast genomes, genetic diversity, phylogeny, *Prunus* subgenus *Cerasus*

## Abstract

Cherries (*Prunus* Subgenus *Cerasus*) have economic value and ecological significance, yet their phylogeny, geographic origin, timing, and dispersal patterns remain challenging to understand. To fill this gap, we conducted a comprehensive analysis of the complete chloroplast genomes of 54 subg. *Cerasus* individuals, along with 36 additional genomes from the NCBI database, resulting in a total of 90 genomes for comparative analysis. The chloroplast genomes of subg. *Cerasus* exhibited varying sizes and consisted of 129 genes, including protein-coding, transfer RNA, and ribosomIal RNA genes. Genomic variation was investigated through InDels and SNPs, showcasing distribution patterns and impact levels. A comparative analysis of chloroplast genome boundaries highlighted variations in inverted repeat (IR) regions among *Cerasus* and other *Prunus* species. Phylogeny based on whole-chloroplast genome sequences supported the division of *Prunus* into three subgenera, I subg. *Padus*, II subg. *Prunus* and III subg. *Cerasus*. The subg. *Cerasus* was subdivided into seven lineages (IIIa to IIIg), which matched roughly to taxonomic sections. The subg. *Padus* first diverged 51.42 Mya, followed by the separation of subg. *Cerasus* from subg. *Prunus* 39.27 Mya. The subg. *Cerasus* started diversification at 15.01 Mya, coinciding with geological and climatic changes, including the uplift of the Qinghai–Tibet Plateau and global cooling. The Himalayans were the refuge of cherries, from which a few species reached Europe through westward migration and another species reached North America through northeastward migration. The mainstage of cherry evolution was on the Qing–Tibet Plateau and later East China and Japan as well. These findings strengthen our understanding of the evolution of cherry and provide valuable insights into the conservation and sustainable utilization of cherry’s genetic resources.

## 1. Introduction

Cherries are members of the *Prunus* subgenus *Cerasus* (Mill.) A. Gray, which is predominantly distributed in the temperate and subtropical regions of the northern hemisphere [1,2]. It encompasses 50–60 species with high economic values owing to their ornamental flowers and edible fruits, as well as serving as valuable resources for cherry genetic improvement [1,3]. Nonetheless, the phylogeny, geographic origin, timing, and biogeography of subg. *Cerasus* species have not been explored. Close genetic relationships among species and frequent interspecific hybridization [4,5,6] are believed to be major obstacles to addressing these uncertainties.

Early attempts to classify and determine the phylogeny of subg. *Cerasus* primarily relied on the morphological features of leaf, inflorescence, flower, and fruit [3,7,8]. However, due to the extensive overlap in the distribution of species and hysteranthy, it is very difficult to discriminate interspecific differentiations from intraspecific variations [9,10,11], and the circumscription of species in this subgenus remains a contentious issue. Even the subdivision within the subgenus based on morphology needs to be seriously tested using molecular data.

With advances in molecular biology and sequencing technology, subg. *Cerasus* has undergone some phylogenetic studies using molecular markers, gene fragments, and even the complete chloroplast (cp) genome [6,12,13,14,15,16,17,18]. Unfortunately, one of the two clades revealed by previous studies consists of only one species (*P. mahaleb*), with all other species in the other clade [12,14,15,16]. The phylogenetic relationships of *Cerasus* species are actually unresolved due to low resolutions of the markers used [19,20,21,22]. The phylogenetic reconstruction of woody plants with a long life history or a short evolutionary history, such as cherries, requires molecular markers of more informative sites or markers of high resolution. Due to technical limits, phylogeny based on whole cp genomes was made affordable only recently. Another limitation of previous molecular systematics of the subg. *Cerasus* is that many species were absent from the studies. This is probably because of sampling challenges due to vast but scattered distribution and specimen-identification difficulties because of the hysteranthy of most species.

The subg. *Cerasus* displays a notable geographic distribution pattern characterized by a disjunct distribution between Eurasia and North America, as well as the Sino-Japanese floristic region. Chin et al. [15] postulated that the ancestor of the subg. *Cerasus* expanded from East Asia to the western part of North America during the Paleocene–Eocene Thermal Maximum (PETM) and Early Eocene Climatic Optimum (EECO) through the ice-free Bering Land Bridge. However, other biogeographical studies suggested that the most closely related species with disjunct distribution likely migrated from East Asia to North America during the Miocene Epoch [23]. The time when the migration of subg. *Cerasus* plants between East Asia and North America happened remains to be estimated. Moreover, the formation of the Sino-Japanese disjunct distribution seems more complex due to diverse terrain and climatic conditions, as well as the complex geological and climatic history and sea-level changes the plants in this region experienced [24,25]. These factors are critical driving forces for the formation of the geographic distribution pattern of species [26]. About 90% of the cherry species occur in the Sino-Japanese floristic region, and the biogeographical relationships of cherry species in this region have drawn a lot of attention.

In this study, we determined 54 cp genomes and downloaded other 36 cp genomes of the *Prunus* species from the National Center for Biotechnology Information (NCBI) database, representing almost all cherry species. Comprehensive analyses of the cp genome data of *Prunus* species were conducted to (1) explore sequence variations in the cp genomes, (2) reveal the phylogenetic relationships among cherry species, (3) estimate the divergence times of lineages within the subg. *Cerasus*, and (4) elucidate the geographic relationships among cherry species.

## 2. Results

### 2.1. Chloroplast Genome Features

We sequenced and de novo assembled 54 cp genomes of *Prunus* species (Appendix A). Furthermore, we incorporated 36 additional cp genomes from the NCBI database, resulting in a total of 90 cp genomes (Appendix A), representing 55 species of subg. *Cerasus*, thirteen species (nine species of subg. *Prunus*, three species of subg. *Padus*) of other subgenera, and two outgroups (one species of *Malus* and one species of *Spiraea*).

The cp genomes of subg. *Cerasus* exhibited a range of size variations, from 157,458 base pairs (bp) to 158,138 bp, with GC content ranging from 36.6% to 36.9% (Figure 1, Appendix A). These genomes had a typical quadripartite structure consisting of a large single-copy (LSC, 85,567–86,261 bp) and a small single-copy (SSC, 19,046–19,253 bp) region that were separated by a pair of inverted repeats (Irs, 26,351–26,468 bp) (Appendix A). The GC contents of the LSC, SSC, and IRs were 33.6–34.6%, 30.0–30.8%, and 42.5–42.6%, respectively (Appendix A).

In the assembled accessions of subg. *Cerasus*, the synteny and gene number were found to be highly conserved. Gene annotation revealed a total of 129 genes in all assembled cp genomes, among which 95 genes were single-copy genes and 17 were multi-copy genes, including 84 protein-coding genes, 37 transfer ribonucleic acid (tRNA) genes, and 8 ribosomal ribonucleic (rRNA) genes (Appendix A). Introns were found in 18 annotated genes in all species, with two introns for *ycf3* and *clpP* and one intron for the other 16 genes (*atpF*, *ndhA*, *ndhB*, *rpoC1*, *rps12*, *rps16*, *rpl2*, *rpl16*, *petB*, *petD*, *trnK*-UUU, *trnG*-UCC, *trnL*-UAA, *trnV*-UAC, *trnI*-GAU, and *trnA*-UGC) (Appendix A). Within the IR region, there were six protein-coding genes, four rRNA genes, and seven tRNA genes. The SSC region contained 12 protein-coding genes and 1 tRNA gene. The LSC region consisted of 82 genes (excluding *rps12*), including 60 protein-coding genes and 22 tRNA genes. Notably, *rps12* had two introns, but with trans-splicing, where one of its exons was located in the LSC region and the other was duplicated by the IRs.

### 2.2. Chloroplast Microsatellite, InDel, SNP, and Genome Structure Variation

The simple sequence repeats (SSR or microsatellites) of the cp genome that display high variability in copy numbers serve as valuable markers for investigating phylogenetic relationships and the biogeography of related taxa. In this study, MISA analyses revealed 7593 microsatellite repeats in 87 *Prunus* plastid genomes (Appendix A). The commonest motifs were mononucleotides (60.22~72.34%), followed by dinucleotides (14.44~23.68%), tetranucleotides (9.57~14.12%), hexanucleotides (0~3.61%), pentanucleotides (0~2.41%), and trinucleotides (0~2.22%). Dinucleotide SSRs contained 79 AG/CT and 1284 AT/AT. Interestingly, ACT/AGT-type trinucleotide SSRs were only present in the cp genomes of *P. humilis*, *P. padus*, and *P. brachypoda*, while ACAT/ATGT-type tetranucleotide SSRs and AAAAG/CTTTT-type pentanucleotide SSRs were only identified in the cp genomes of the subg. *Padus* (Appendix A).

To investigate mutational hotspots in *Prunus* plastomes, we conducted an analysis of insertions/deletions (InDels) and single-nucleotide polymorphisms (SNPs) by mapping the 86 *Prunus* cp genomes to the reference genome of *P. cerasoides* (no. MF621234) (Appendix A). Our examination of the *Prunus* cp genomes revealed that InDels and SNPs were primarily distributed in the upstream and downstream regions of protein-coding genes (Appendix A). Overall, we identified a total of 2928 InDel loci and 5424 SNP loci, with effect numbers of 23,566 and 41,999, respectively (Appendix A). For InDel variants, the insertions exceeded the deletions by 662, with up to 16 allele types and 85 bp in length in one locus. Furthermore, the impact level with the highest count was the modifier, with 23,361 occurrences, accounting for approximately 99.13% of all impact levels (Appendix A). Among the identified SNP loci, the most common nucleotide substitution across the taxonomic groups was the transition from G to A, whereas the transversion mutation from C to G was the rarest (Appendix A). Remarkably, out of a total of 1924 SNPs, there were 883 non-synonymous SNPs, making up 45.89% of the total (Appendix A). Among these non-synonymous SNPs, 878 were missense mutations, while five were nonsense mutations. Furthermore, when considering all cp genes, it was noteworthy that *rps12* exhibited a notably higher frequency of SNPs, while *ycf1* accumulated the highest number of mutations at the high and moderate levels (Appendix A).

To elucidate the contraction and expansion of the IR regions among subg. *Cerasus* species and other *Prunus* species, comparative analyses of the boundaries between the LSC region and the IR regions, as well as analyses of the boundaries between the IR regions and the SSC region, were performed and are illustrated in Appendix A. In comparison to other *Prunus* species, IR regions in the subg. *Cerasus* species exhibited varying degrees of contraction and expansion. For instance, *rps19* had a segment located in the IRb region with a length ranging from 175 to 217 bp, while another segment was positioned in the LSC region, spanning from 62 to 104 bp. As a result, this gene was not completely duplicated, and the replicated lengths differed among different species. The *ndhF* gene commonly underwent variations at the boundary between the IRb and SSC regions. In species of the subg. *Padus* and subgenus *Prunus*, *ndhF* spanned both the SSC and IRb regions, with a portion of 4 to 20 bp located within the IRb region. However, in the subg. *Cerasus* species, *ndhF* was predominantly found within the SSC region, positioned 1 to 28 bp away from the IRb region. Except for *P. mahaleb*, *P. setulosa*, and *P. tianshanica*, where *ycf1* was entirely located in the IRa regions, the *ycf1* in other species spanned both the IRa and SSC regions, resulting in incomplete duplication and the formation of a pseudogene (Appendix A).

In order to explore potential DNA barcodes, we screened 75 cp genomes from 55 species of subg. *Cerasus* for hypervariable regions. By using multiple sequence alignment, we identified the following hypervariable regions: *matK*-*rps16*, *petN*-*psbM*, *psbD*, *trnF*-GAA-*ndhJ*, *atpB*-*rbcL*, *psbE*-*petL*, *psbB*, *ndhE*, *ndhE*-*ndhG*, *trnN*-GUU-*trnR*-ACG, and *trnR*-ACG-*rrn5* (Appendix A). A total of 312 variable sites were identified in these 11 regions, and the nucleotide diversity per site (Pi) ranged from 0.0045 to 0.0058 (Appendix A). These results revealed the existence of mutations in the hypervariable regions of the consistently stable chloroplast genome, which could be potential candidates for developing DNA barcodes for molecular delimitation of subg. *Cerasus*.

### 2.3. Phylogenetic Relationships among Lineages

The aligned dataset of 90 cp genomes contained 856 variable sites and 384 parsimony-informative sites (Appendix A, Appendix A). Trees constructed using maximum-likelihood (ML) and Bayesian inference (BI) methods had identical topologies (Appendix A). The phylogeny revealed three strongly supported lineages (I, II, and III) (Figure 2), with confidence levels of 100% bootstrap support (BS) and 1.00 posterior probability (PP). Lineages I and II comprised species of subg. *Padus* and subg. *Prunus*, respectively, while Lineage III comprised species of subg. *Cerasus*, indicating monophyly of the subgenera. Lineage III had diverged into seven sublineages numbered from IIIa to IIIg according to the branch lengths and BS and PP values. Although structural analysis suggested three groups (K = 3) in subg. *Cerasus* (Appendix A, Appendix A), the phylogenetic trees did not show remarkable differences between IIIb and IIIc-IIIg.

The basal sublineage IIIa was the SW Asian and European species *P. mahaleb* with corymbose–racemose inflorescences. Sublineage IIIb consisted of six species occurring in Southwest China (Figure 2). Sublineage IIIc consisted of species in the Far East, North America, and West China. There are three small groups of species in this sublineage: (1) the far eastern *P. maximowiczii* and its North American alliance; (2) Sino-Japan island element *P. subhirtella* and its variants; and (3) seven western China elements. Sublineage IIId included four Eurasian cherries, *P. avium*, *P. cerasus*, and *P. fruticosa*, and a species in western China, *P. setulosa*. Sublineage IIIe comprised taxa mostly occurring in southwestern and central China. Sublineage IIIf consisted of three species endemic to western, central and southwestern China. Sublineage IIIg is the largest group encompassed taxa distributed in eastern China (four species), Taiwan Island (three species), Japan (eight species) and the Korean Peninsula (three species).

### 2.4. Divergence Time

The estimation of divergence times provided valuable insights into the evolutionary history of *Prunus*. Molecular dating in this study indicated that subg. *Padus* diverged from subg. *Cerasus* and subg. *Prunus* in the early Eocene approximately 51.42 million years ago (Mya), with a 95% highest posterior density (HPD) ranging from 40.13 to 64.32 Mya. The subg. *Cerasus* separated from subg. *Prunus* in the middle Eocene around 39.27 Mya (95% HPD: 28.90–49.08 Mya) (Figure 3). However, the species in subg. *Cerasus* are much younger. The first divergence event within subg. *Cerasus* happened in the middle Miocene about 15.01 Mya (95% HPD: 13.27–16.74 Mya) when *P. mahaleb* diverged from the common ancestor (Figure 3). Soon, the lineage IIIb diverged 13.31 Mya (95% HPD: 11.20–15.45 Mya) and so forth. Noteworthily, (1) the Far East/North America taxa diverged from their sister members 9.71 Mya (95% HPD: 7.10–12.18 Mya, IIIc); (2) the eastern Chinese–Japanese–Korean members (IIIg) diverged from their close relatives (IIIf) 7.57 Mya (95% HPD: 5.06–10.02 Mya) (Figure 3).

## 3. Discussion

The subg. *Cerasus* comprises a diverse array of plant species that hold significant value in terms of their ornamental and economic attributes. Despite their potential, the majority of cherry species remain in their wild state and have not been subjected to extensive exploitation and cultivation except in Japan [27]. In contrast to their cultivated counterparts, wild cherry species have undergone prolonged adaptation to the changing environmental conditions, equipping them with resilient mechanisms to sustain both biotic and abiotic stresses. The evolutionary journey has facilitated the accumulation of extensive genetic variations and the acquisition of valuable gene reservoirs within wild populations. These genetic resources assume a pivotal role in genetic-enhancement programs, enabling the targeted selection of desirable traits [28]. However, to ensure the effective preservation and utilization of germplasm resources, a fundamental prerequisite lies in unraveling the phylogenetic relationships that govern the intricate web of species within the subg. *Cerasus*. In this study, we conducted comprehensive phylogenetic analyses of the subg. *Cerasus* by integrating a dataset consisting of 54 newly assembled cp genomes of subg. *Cerasus* and its closely related taxa, along with 36 publicly accessible cp genomes, representing 45 species in mainland China, 5 species in island Taiwan, 11 species in Japan, 6 species in the Korean Peninsula, 6 species in Russia, 4 species in Europe, 4 species in South Asia, 3 species in West Asia, 2 species in Central Asia, and 2 species in North America [3,7,29,30,31,32,33]. This dataset encompassed nearly all known species within subg. *Cerasus*, providing a robust foundation for our investigation.

### 3.1. Phylogenetic Relationship within Subg. Cerasus

In our phylogenetic trees, the subgenus *Cerasus* was composed of seven distinct subclades, referred to as subclades IIIa to IIIg for the first time. The basal position of the subclade IIIa with a single species, *P. mahaleb*, is consistent with previous studies [12,14,15,16] and supports the notion that in *Prunus* raceme is a plesiomorphic trait [5,18]. *P. mahaleb* bridges the gap between the basal subg. *Padus* with raceme and subg. *Cerasus* mostly with cymose. The second breakthrough is the determination of the earliest divergence of six Himalayan cherry species (IIIb). The Himalayans are the cradle and divergence center of cherry species. The third one is that the phylogenetic relationship between two North American species, *P. pensylvanica* and *P. emarginata*, and the Eurasian cherry species has been resolved. The North Asian *P. maximowiczii* is more closely related to the North American *P. pensylvanica* and *P. emarginata*, suggesting links among North American, North Asian, and Himalayan cherry species in IIIc.

In previous classification studies, the central Asian and European *P. fruticosa*, *P. Cerasus*, and *P. avium* were considered closely related and placed in sect. *Cerasus* [8,34,35], with *P. cerasus* being a hybrid between *P. fruticosa* and *P. avium* [36]. Their close genetic relationships were confirmed in this study and formed a clade together with *P. setulosa* (IIId). Although the Chinese *P. setulosa* and the European *P. avium* share a common ancestor, they have diverged for a long time, judging from long stem branch lengths. Clade IIIe and clade IIIf consist of species endemic to China.

Clade IIIg comprises species predominantly occurring in eastern China, Taiwan Island, Japan, and the Korean Peninsula. Previous studies have suggested morphological similarities among *P. serrulataerrulate*, *P. jamasakura*, and *P. leveilleana*, leading to their classification as the *P. serrulataerrulate* complex [11]. However, this study did not indicate a close phylogenetic relationship among them. Instead, *P. serrulata*, *P. sargentii*, *P. takesimensis*, *P. kumanoensis*, and *P. serrulata* var. *lannesiana* formed a clade, while *P. jamasakura* and *P. leveilleana* exhibited a closer phylogenetic relationship with *P. apetala*, *P. incisa*, *P. nipponica*, and *P. speciosa*. Notably, *P. takesimensis*, which was found on Ulleung Island in the Korean Peninsula, shares morphological similarities with *P. sargentii*, suggesting that *P. sargentii* may serve as the continental progenitor of *P. takesimensis* [37]. However, our results indicated that *P. takesimensis* had significantly diverged from *P. sargentii* and *P. serrulata* var. *lannesiana*. According to the flora of Japan, *P. serrulata* var. *lannesiana* is a cultivated variety, possibly derived from *P. speciosa*, *P. jamasakura*, and *P. sargentii* [38]. Our phylogenetic tree suggested that *P. serrulataampanula* var. *lannesiana* is indeed closely related to *P. sargentii*. Furthermore, three species in Taiwan Island, namely *P. takasagomontana*, *P. transarisanensis*, and *P. matuurai*, formed a clade, indicating their close phylogenetic relationship and distinctness from other sympatric cherry species with (*P. campanulata* and *P. dielsiana*). The species endemic to Japan, *P. speciosa*, *P. jamasakura*, *P. nipponica*, *P. apetala*, *P. incisa*, and *P. leveilleana*, are natural except *P. jamasakura*, which is very similar to *P. serrulata* var. *lannesiana* morphologically. Therefore, the systematic position of both *P. serrulata* and *P. jamasakura* remains to be determined.

### 3.2. Origin and Dispersal of Subg. Cerasus Species

The genus *Prunus* comprises three subgenera: subg. *Padus*, subg. *Cerasus*, and subg. *Prunus*. Based on the morphological characteristics of their inflorescences, these subgenera correspond to racemose, corymbose, and solitary inflorescences, respectively. According to Chin et al., the subgenera have diverged for 54 to 56 Mya [15]. The divergence time of subg. *Padus* is roughly confirmed in this study at 51.42 Mya. Around 51–53 Mya, during the Early Eocene, there was a period of sustained warm climate known as the early Eocene climatic optimum [39]. This warm climate may have facilitated the early differentiation of the genus *Prunus*. However, subg. *Cerasus* and subg. *Prunus* were estimated to be much younger in this study, about 39.27 Mya than the estimate by Chin et al., but much older than the estimate by Zhang et al. [14]. It has been suggested that the Eocene–Oligocene transition (EOT) around 30–40 Mya marked a period of global cooling that resulted in significant flora turnovers [40,41,42].

Although subg. *Cerasus* started diverging from subg. *Prunus* around 39.27 Mya, no species has survived today until 15.01 Mya. Now, cherries are common trees in the Northern Hemisphere (Figure 4a). Based on the Bayes–DIVA analysis, the ancestral distribution areas were western China and Central Asia (Figure 4b), especially the Himalayan region [15,28]. Ancestral cherries took several routes to conquer the Northern Hemisphere (Figure 4c) besides the in situ diversifications in the Himalayan area. The first one was taken by the relic *P. mahaleb* from the refuge in the mid-Miocene climatic optimum (MMCO, 15–17 Mya) through Central Asia and western Asia into Europe. It was suggested that plants in the Himalayan region spread from East Asia to Europe during the Oligocene, with the Qinghai–Tibet Plateau (QTP) and the lowlands of the Himalayas serving as corridors. However, the rise of the QTP during the Miocene led to the disappearance of this migration route [43]. The pathway through Central Asia to Europe was facilitated by the closure of the Turgai Strait in the Middle Eocene (∼29 Mya), allowing East Asian plants to enter Europe via Central Asia. Furthermore, under the warm and humid climate during the MMCO and the enhanced Asian summer monsoon [44], the ancestor of the subg. *Cerasus* migrated from high-altitude regions in the west to temperate deciduous forests and warm-temperate evergreen forests in the eastern altitudinal zone. The first differentiation occurred around 15 Mya and coincided with the rapid uplift of the QTP and the onset of global cooling and aridification [39,45,46], indicating significant geological and climatic changes during this period that drove subsequent diversifications of cherry species.

In situ diversifications are well exemplified by the species in IIIb. The species in this clade is mainly distributed in the southern and southeastern parts of the QTP, specifically the Himalayans and the Hengduan Mountains (HHM) (Figure 4b). Their differentiation times ranged from 11.2 Mya to 2.83 Mya, indicating constant speciation in this area before the onset of the Quaternary glaciation. The disjunct biogeographical relationships between East Asia and North America are exemplified by species in IIIc. The species in this clade had diverged into three groups about 9.71–9.27 Mya (Figure 4b,c). The first group migrated southeastward to central to western China and underwent in situ diversification, forming seven species. The second group migrated into the Korean Peninsula, Japan, and Taiwan Island, substantially remaining one or two species. And the third group migrated to Northeast China, the Far East of Russia, and North America through the Bering Land Bridge and diverged into three species, with one (*P. maximowiczii*) in Northeast China and the Far East of Russia and two (*P. emarginata* and *P. pensylvanica*) in North America. The time for cherries to reach Eastern North America is quite similar to that for other species [23,47].

The floristic relationships between Asian and European cherries are represented by species in clade IIId besides the relic species *P. mahaleb*. This is a small clade with three species (excluding hybrid *P. cerasus*), including one species (*P. setulosa*) in China and two species (*P. avium* and *P. fruticosa*) from Central Asia to Europe. They have diverged for 6.81–9.07 Mya (Figure 4b). Similarly to IIIc, *P. setulosa* migrated southeastward into northwestern China, and *P. avium* and *P. fruticosa* migrated westward from Asia to Europe (Figure 4c). Unlike American cherries, which diverged into two species after reaching North America, the two central Asian and European cherries diverged in Asia and then migrated to Europe instead. The Chinese *P. setulosa* and the central and European *P. avium* are sibling species. *Prunus fruticosa* diverged earlier.

Western China was the mainstage of cherry speciation after the uplift of QTP. However, with the intensification of the East Asian monsoon at 3.6 Mya, central and eastern China became the mainstages of species formation. The most typical example is the species on the second gradient terrains of China (IIIe and IIIf). In IIIe, their ancestor diverged 8.89 Mya, but species burst happened after 3.08 Mya. The intensified East Asian monsoon at 3.6 Mya may have promoted their differentiation [48,49], especially for cherries in eastern China, Taiwan Island, Japan, and the Korean Peninsula (IIIg). Undoubtedly, cherry species in Taiwan Island, Japan, and the Korean Peninsula in this clade were from China (Figure 4c). Although it was believed that the Japanese archipelago began to be separated from the Eurasian continent around 24 Mya, the two regions had repeated connections through land bridges, including the East China Sea Land Bridge, the Korea Strait Land Bridge, and the Taiwan Land Bridge, since the middle Miocene, owing to sea-level fluctuations caused by climate changes [50,51,52]. According to the divergence times of species groups in IIIg, their ancestors had already reached Taiwan Island, Japan, and the Korean Peninsula before the Pleistocene. The rising sea levels caused the previously connected populations to become isolated, which has been an important driving force for population differentiation and speciation [24,52,53].

According to the differentiation times shown in Figure 4b, 59.3% of the species and varieties in the subg. *Cerasus* originated from climate-driven vicariance during the Pleistocene and were mainly distributed in the Sino-Japan forest subkingdom. This indicates that the climatic changes during the Quaternary had a substantial impact on the speciation of the cherry species in this region. In contrast, species that originated before the Pleistocene were mostly distributed in the Sino-Himalayan forest subkingdom, suggesting that the uplift of the QTP and the associated geological and climatic changes were important drivers for the speciation of the cherry plants in this region.

### 3.3. Taxonomic Implications

The modern taxonomic subdivision of a taxon is evidence-based, especially on phylogenetic relationships. Even so, the grouping of clades into formal taxonomic rank is actually optional to taxonomists. The phylogeny of subg. *Cerasus* (Figure 2) provides a backbone for sectional subdivision, and the division of clades is believed to be helpful to determine sections. For example, IIIa matches sect. *Mahaleb* (Koehne) Yu et Li, IIId matches sect. *Cerasus*, and IIIe roughly matches sect. *Lobopetalum* (Koehne) Yu et Li [1]. The seven-clade subdivision does not completely fit the existing taxonomic sections but agrees with geographical distribution patterns. It is probably because the phylogeny was based on maternally transmitting cp genomes.

About half of cherry species have evolutionary histories of a few million years. Such species are too young to build strict reproductive isolation mechanisms. Therefore, interspecific hybridization happens occasionally where they meet. According to the biological concept of species, they are semispecies, formerly known as subspecies. However, such instances are very common in East Asian flora, and most of them were treated as distinct species. In subg. *Cerasus*, there are some typical examples. The species *P. itosakura*, *P. pendula*, *P. spachiana* f. *ascendens*, *P. subhirtella*, and *P. taiwaniana* in IIIc are closely related and are better treated as one species. Some Japanese cherries in IIIg are vicarious species. Similarly, the species status of cherries in IIIe is open to question.

It seems premature to make a formal taxonomic revision of subg. *Cerasus* either at the sectional or at species ranks before a reliable phylogeny based on nuclear genomes (or genes) is available. Given this, we intend to employ nuclear genome sequences for identifying potential hybridization events among subg. *Cerasus* taxa. On this basis, we aim to construct a more detailed backbone phylogeny of subg. *Cerasus* through comparison with the outcomes of this study in future research.

## 4. Materials and Methods

### 4.1. Plant Material and Taxon Sampling

A total of 90 cp genomes were included in our study, representing all three subgenera of *Prunus sensu lato* [16] and three outgroup species (*Malus domestica*, *M. micromalus*, *Spiraea martini*), among which 54 genomes were determined in this study and 36 genomes were downloaded from the NCBI database. The 87 ingroup represented 55 species of subg. *Cerasus* (75 cp genomes), 9 species from subg. *Prunus*, and three 3 from subg. *Padus*. The samples used in this study for cp genome sequencing were collected from the field (47 samples) or obtained from DNA Bank of China (7 samples).

### 4.2. DNA Extraction and Sequencing

Total genomic DNA was extracted from the silica-gel-dried leaf materials using a modified cetyltrimethylammonium bromide (mCTAB) protocol [54] and purified using the Wizard DNA Clean-Up System (A7280, Promega Corporation, Madison, WI, USA). DNA concentration was assessed using the Qubit 2.0 Fluorometer (Thermo Fisher Scientific, Waltham, MA, USA). Agarose gel electrophoresis was employed to quantitatively determine the lengths of DNA fragments. The genomic DNA was used to construct a short-insert (<800 bp) paired-end sequencing library. Paired-end sequencing was performed on the HiSeq X Ten analyzer (Illumina, San Diego, CA, USA) at Novogene Co., Ltd. (Tianjin, China). The sequencing process yielded a large volume of data for most samples: approximately 10 gigabases (Gb) of 150 bp paired-end reads each sample.

### 4.3. Chloroplast Genome Assembly and Annotation

Each raw data set was cleaned and filtered with Trimmomatic v0.39 [55]. Clean reads were assembled de novo into contigs with SPAdes v3.9 [56]. The cp genome contigs were screened in Blastn v2.8.10 [57] using the close-related and published species of *P. cerasoides* (no. MF621234) [58] as a reference. The extracted contigs were preliminary assembled with Sequencher v5.4.5, and the gaps between contigs were filled with clean reads. The cp genomes were further assembled and checked by mapping against the original reads. Subsequently, manual curation and validation were performed to ensure the precision and fidelity of the assembled cp genome using Geneious R10.2.3 [59]. The cp genome annotation and correction were conducted using Plann v1.1.2 and Sequin [60]. The genome map was drawn using Circos v0.69-9 [61].

### 4.4. Chloroplast Genome Alignment and Polymorphism Assessment

The whole-genome dataset was divided into a series of datasets using Geneious R10.2.3—including the protein-coding gene dataset, intergenic region dataset, LSC region dataset, SSC region dataset, and IR region dataset—for variation analyses. Each dataset was aligned with MAFFT v7.408 [62] and adjusted manually with Se-Al v2.0 a11 [63] if necessary. The complete cp genomes of *Prunus* were compared using mVISTA software with the Shuffle-LAGAN mode (https://genome.lbl.gov/vista/mvista/submit.shtml (accessed on 15 March 2023)) [64] against the reference *P. cerasoides* (no. MF621234). Comparative analysis of genes near boundaries of LSC/IRb/SSC/IRa regions in cp genomes was executed using the IRscope software (https://irscope.shinyapps.io/irapp (accessed on 26 March 2023)) [65].

A total of 84 protein-coding genes were extracted from the cp genome of the subg. *Cerasus* dataset. Additionally, 78 unique protein-coding genes were used for subsequent variation analyses after removing duplicate sequences. We employed DnaSP v6.12.03 [66] and MEGA v7.0.26 [67] to analyze the nucleotide polymorphisms of different datasets.

### 4.5. Genomic Variation Analyses

We aligned *Prunus* cp genome sequences to *P. cerasoides* (no. MF621234) using nucmer (parameters: -mum-g 1000-c 90-| 40-t 20) in the MUMmer v4.0.0 [68]. The alignment table was filtered with delta-filter (parameters: −1-i 90-| 40). The filtered alignments were then transformed into a readable format using show-coords (parameters: -THrd). Using show-coords outputs, we identified SNPs and InDels using SyRi v1.6.3 with default parameters except –no-chrmatch [69]. Independent SNP and InDel files obtained from different individuals were transformed and merged into a unified variant call format (vcf) file using BCFTools v1.9 [70]. To assess the distribution patterns and potential effects of SNPs and InDels across the cp genomes, we utilized BEDTools v2.27.0 [71] and SnpEff v4.2 [72].

Simple sequence repeats (SSRs) in each cp genome were screened using MISA software (https://webblast.ipk-gatersleben.de/misa (accessed on 5 April 2023)) [73] with the following parameters: at least 10 repeat units for mononucleotides, 5 repeat units for dinucleotides, 4 repeat units for trinucleotides, and 3 repeat units for tetra-, penta-, and hexanucleotides. In addition, the maximal number of interrupting bases between two SSRs in a compound microsatellite was 100 bases.

In order to quantify selective pressure, we utilized EasyCodeML v1.0 [74] to compare the rates of non-synonymous (dN) and synonymous (dS) substitutions. This analysis allowed us to assess the impact of selective pressures on the genes of interest. We calculated the ratios (ω) of dN/dS, where ω = 1, ω > 1, and 0 < ω < 1 represented neutral, positive, and purifying selection, respectively. We used the species phylogenetic tree obtained via reconstruction as the input tree file and conducted a site model to detect selection (Appendix A).

### 4.6. Phylogenetic Analyses

To ensure the accuracy of subsequent phylogenetic analyses, we assessed the nucleotide substitution saturation of the whole-genome dataset using DAMBE v7.3.2 [75] and the phylogenetic signal using IQ-TREE v1.6.12 [76]. We selected the best-fit models using ModelFinder for each phylogenetic analysis based on the Bayesian information criterion (BIC) [77]. We utilized ML methods to generate phylogenetic trees for the whole-genome dataset using the IQ-TREE plugin in PhyloSuite v1.2.3 with 10,000 regular bootstrap replicates [78]. BI analyses were performed with MrBayes [79] on XSEDE v3.2.6 in CIPRES Science Gateway V 3.3 [80], based on the best-fit model GTR + F + I + G4, Nst = 6, rates = invgamma, using a Markov chain Monte Carlo (MCMC) algorithm with 1,000,000,000 generations and sampling every 10,000 generations. PP was calculated from the majority consensus of all the sampled trees when the standard deviation of the split frequencies permanently fell below 0.01, and the first 25% of trees sampled during the burn-in phase were discarded.

The genetic structure was estimated using Structure v2.3.4 [81,82], with the pre-defined genetic clusters (K) ranging from 2 to 7; the analysis was repeated 10 times. The optimal number of K was determined based on the ΔK value calculated using Evanno’s method [83].

### 4.7. Divergence Time Estimation and Ancestral Geographic Distribution Inference

We employed BEAST v1.8.0 [84] to estimate the divergence times of *Prunus* using the *Prunus* cp genome dataset. Four calibration priors from the Paleobiology database (http://paleobiodb.org accessed on 10 April 2023) were utilized in this study (Appendix A): (1) a fossilized *P. wutuensis* endocarp discovered in Shandong province, China, was used to calibrate the divergence of the most recent common ancestor of the subg. *Cerasus*, subg. *Prunus*, and subg. *Padus*, with a mean of 55 Mya [85]; (2) a *P. triloba* fossil was used to calibrate the divergence of the sect. *Amygdalus* within the subg. *Prunus* (offset = 25.72 Mya) [86]; (3) a *P. hirsutipetala* fossil was used to calibrate the divergence of the subg. *Padus* and subg. *Cerasus*, at approximately 33.90–37.20 Mya in the Late Eocene [87]; and (4) a *P. mahaleb* fossil was used as an indirect calibration point [88,89]. We ran analyses with a GTR + Gamma + Invariant Sites model, relaxed clock lognormal to account for rate variation among lineages, Yule tree speciation models, and six independent runs with the MCMC method, each with 300,000,000 generations and sampling every 1000 generations. The adequacy of the parameters was assessed using Tracer v1.7.2 [90] to evaluate convergence and ensure a satisfactory effective sample size (ESS) exceeding 200. Tree files and log files from the six independent runs were combined with LogCombiner after discarding the first 20% of trees as burn-in. A maximum clade credibility tree was generated using TreeAnnotator.

Based on the documented information regarding the native distribution ranges of species within the subg. *Cerasus*, subg. *Prunus*, and subg. *Padus*, along with specimen and sampling data, the distribution area of the subg. *Cerasus* was divided into seven regions, including Region A: Northeast China and the Far East; Region B: North, Northwest China, and Central Asia; Region C: Southwest China; Region D: East, Central, and South China, and Taiwan Island; Region E: Japan and Korean Peninsula; Region F: Europe and West Asia; and Region G: North America. The ancestral geographic distribution reconstruction was conducted using the S-DIVA (Statistical Dispersal-Vicariance Analysis) method in RASP v4.2 software [91], with the same strategy and data matrix as the divergence time estimation.

## Figures and Tables

**Figure 1 ijms-24-15612-f001:**
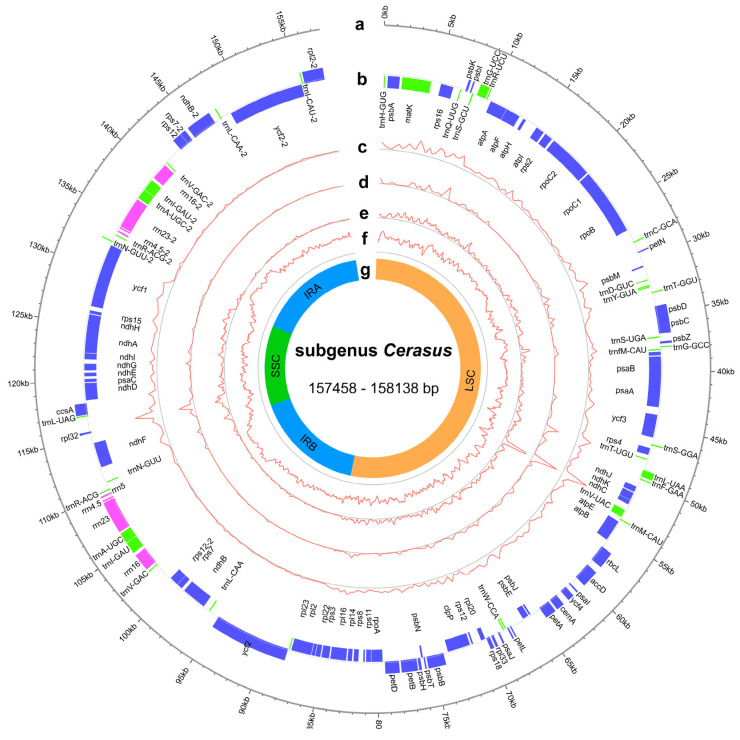
Representative genome map of the subg. *Cerasus* cp genomes. (**a**) Subg. *Cerasus* cp ideogram, scale in kb. (**b**) Gene distribution (blue, protein-coding genes; green, tRNAs; pink, rRNAs). (**c**) Density of single-nucleotide polymorphism (SNP) (400 bp window). (**d**) Density of insertion/deletion (InDel) (400 bp window). (**e**) Nucleotide diversity (Pi) (400-bp window). (**f**) GC content distribution (400 bp window). (**g**) The structure of cp genomes, including LSC region, SSC region, and IR regions.

**Figure 2 ijms-24-15612-f002:**
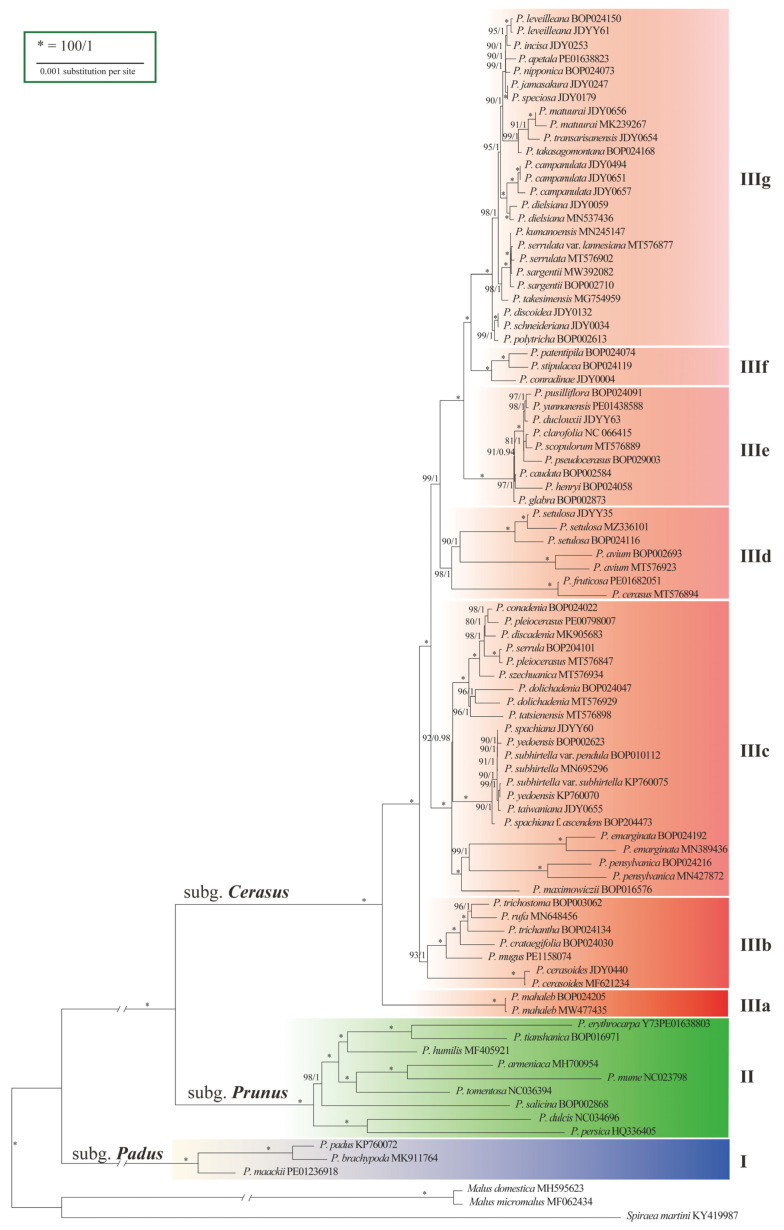
Phylogenetic tree of *Prunus* based on cp genome data, showing divergences within subg. *Cerasus*. Bootstrap supports (BS)/Bayesian posterior probabilities (PP) are shown above the branches.

**Figure 3 ijms-24-15612-f003:**
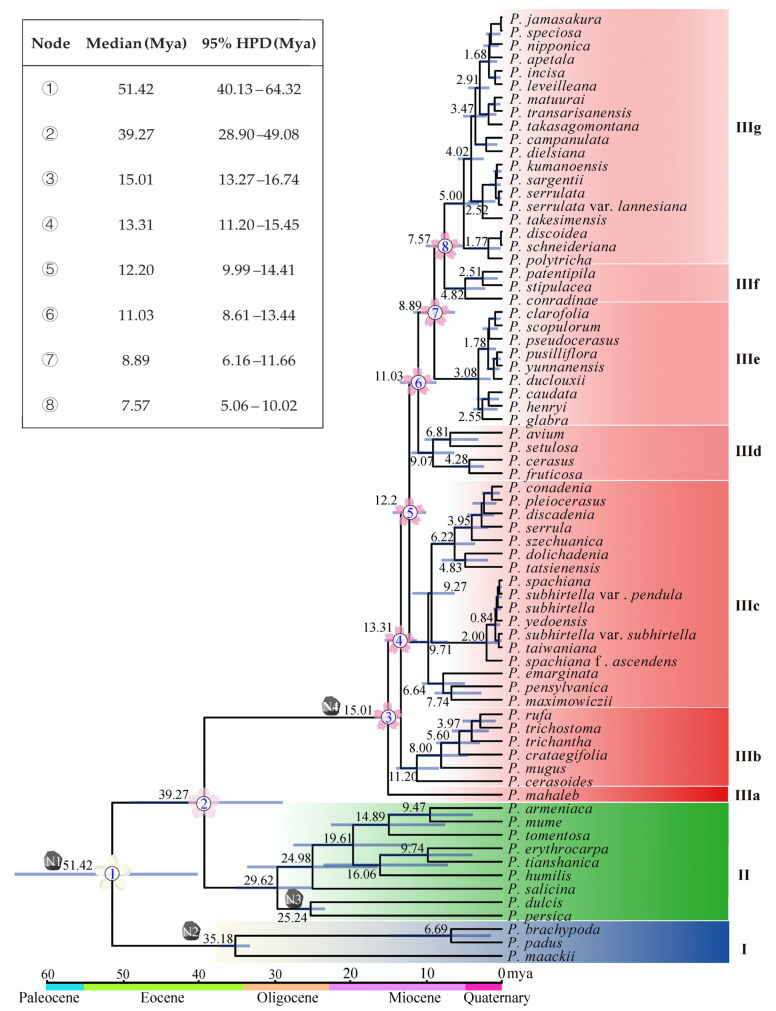
Chronogram of *Prunus* based on cp genome data. N1, N2, N3, and N4 mean the position of the calibration points. Median values and 95% HPD are shown on the branches, with the length of the blue bar indicating the range of HPD. The divergences of subgenera and the clades of subg. *Cerasus* are numbered, and the 95% HPD ranges are given in the upper left corner.

**Figure 4 ijms-24-15612-f004:**
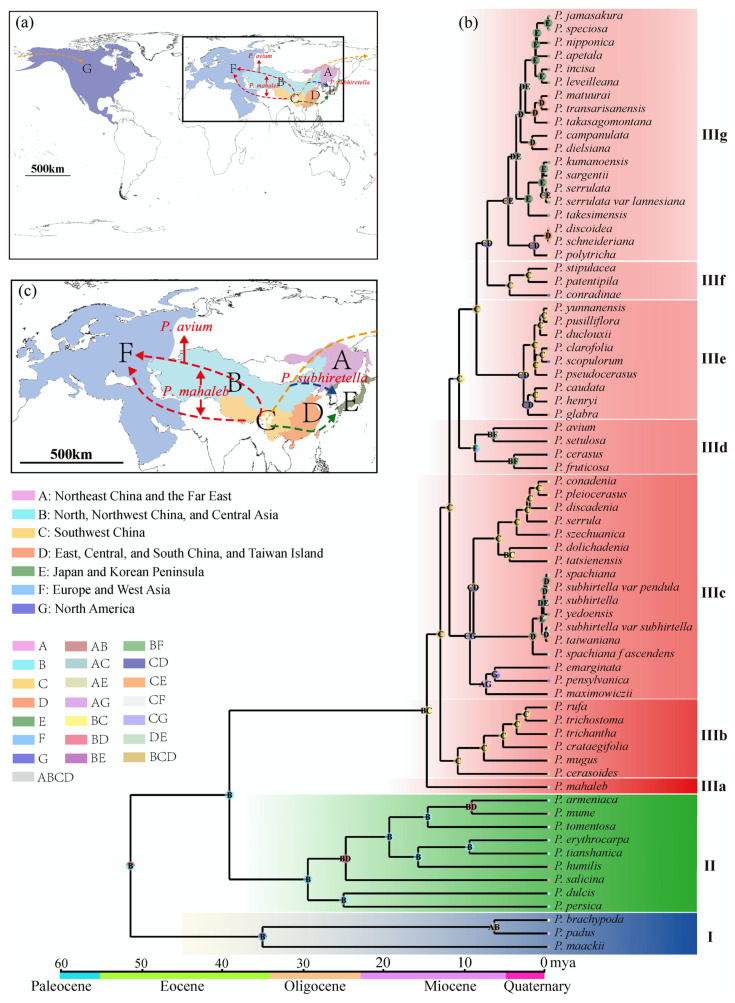
Reconstruction of ancestral areas for *Prunus*. (**a**) Map of possible dispersal routes of the subgenus *Cerasus*. The black square is enlarged for specification in Figure 4c. The capital letters correspond to the ancestral regions as illustrated below. (**b**) Ancestral area reconstructions based on the Statistical-Dispersal–Vicariance-Analysis (S-DIVA) method. Each pie chart at the node represents the frequency of each ancestral area through the proportion of different colors, with the maximum area set to 4. Letters in the pie charts indicate the areas with the highest relative probabilities. (**c**) Enlargement of possible dispersal routes of the subgenus *Cerasus*. Two red dotted lines represent the putative dispersal route of *P. mahaleb*, of which the top line represents the putative dispersal route of *P. avium*; the blue dotted line represents the possible dispersal route of *P. subhiretella*.

## Data Availability

The data presented in this study are available under Genbank accession OR687382-OR687433, ON186529, and ON186527 at NCBI database.

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
