# Peer review of "Evolution of Cherries (Prunus Subgenus Cerasus) Based on Chloroplast Genomes"

_ijms, 2023, doi:10.3390/ijms242115612_

Round 1

Reviewer 1 Report

The manuscript by Shen et al. entitled “Evolution of Cherries (Prunus Subgenus Cerasus) Based on Chloroplast Genomes” analyzes plastome diversity within using novel and previously sequenced genomes. Authors sequenced and assembled 54 chloroplast genomes and performed rather standard comparative analyses. The paper is generally well written and suitable for publication in IJMS after some correction listed below:

Major concerns

  1. I would replace Figure 1 (which didn’t carry any important or new facts) with a figure representing SNP analysis along the plastome. The several data could be visualized, beside SNP (from table S4, S5) - Pi diversity (from Figure S3), diagnostic nucleotides and output from selection analysis.

  2. Table S1 is missing GenBank Ids for newly sequenced plastomes.

  3. There’s no information about sequencing results, plastome coverage and potential heteroplasmic sites

  4. Methodology behind SNP detection (lines 447-452) is not clear - Is the analysis based on raw read mapping or on plastome alignment? What were parameters for SNP detection? How were indels threatened?

  5. It would be nice to see some superbarcoding approach in this paper, because there is no word about the potential of the plastome in molecular delimitation of Prunus.

Minor issues:

  1. Table 1 could be moved to Supplementary, since it doesn't provide any new insight into Prunus plastid gene content.

Line 429: There’s only one map, so this sentence should be singularized.

Author Response

Major concerns

Point 1: I would replace Figure 1 (which didn’t carry any important or new facts) with a figure representing SNP analysis along the plastome. The several data could be visualized, beside SNP (from table S4, S5) - Pi diversity (from Figure S3), diagnostic nucleotides and output from selection analysis.

Response1: Thanks for your suggestion. I have added sequence and variation information to Figure 1, including gene distribution, exon and intron, SNP and InDel density, Pi diversity, GC content, and chloroplast genome structure.

Point 2: Table S1 is missing GenBank Ids for newly sequenced plastomes.

Response2: Thanks for your suggestion. I have submitted the SRA of newly sequenced plastomes to NCBI under the bioproject PRJNA1017114, but these data are processing and have not been released.

Point 3: There’s no information about sequencing results, plastome coverage and potential heteroplasmic sites.

Response3: Yes, I have noticed it, and your suggestion is very helpful. I have provide the information about sequencing results in Table S2.

Point 4: Methodology behind SNP detection (lines 447-452) is not clear - Is the analysis based on raw read mapping or on plastome alignment? What were parameters for SNP detection? How were indels threatened ?

Response4: Thanks for your suggestion. I have revised and provide more details of SNP detection and indel threatened in the methods part.

Point 5: It would be nice to see some superbarcoding approach in this paper, because there is no word about the potential of the plastome in molecular delimitation of Prunus.

Response5: Thanks for your suggestion. For the superbarcoding approach in molecular delimitation of Prunus, we have another paper submitted to study and discuss, so we do not refer to it in this paper.

Minor issues

Point 1: Table 1 could be moved to Supplementary, since it doesn't provide any new insight into Prunus plastid gene content.

Response1: Thanks for your suggestion. I have moved it to Supplementary as Table S4.

Point 2: Line 429: There’s only one map, so this sentence should be singularized.

Response2: Thanks for your suggestion. I have revised it.

Reviewer 2 Report

In the submitted manuscript by Shiliang Zhou, Dongyue Jiangand and colleagues entitled “Evolution of Cherries (Prunus Subgenus Cerasus) Based on Chloroplast Genomes”, the authors conducted chloroplast genome analysis for investigating phylogenetic relationships and genomic variation in subg. Cerasus. Overall, the manuscript is well-written, and the figures have a good presentation. However, there are some issues that should be carefully addressed.

Major and minor comments,

Discussion section

Lines 221-223. Please provide a relative reference.

Lines 227-229. Please provide a relative reference.

Line 392: The authors mention ‘It seems premature to make formal taxonomic revision of subg. Cerasus either at sectional or at species ranks before a reliable phylogeny based on nuclear genomes (or genes) was available.’ I agree with this sentence. So, you should close the section with a perspective about your future work (genome phylogenetic analysis).

MM section

Line 400. The authors mention ‘The 87 ingroup represented 55 species of subg. Cerasus, nine species from subg. Prunus, and three species from subg. Padus.’ Total 55+9+3= 67 form 87, 20 is missing, please incorporate the 20 that are missing. For example, 55 species of subg. Cerasus (75 cp genomes).

Line 402. The authors mention ‘For subg. Cerasus species, there were 45 species distributed in mainland China, five species in island Taiwan, 11 species in Japan, six species in the Korean Peninsula, six species in Russia, four species in Europe, four species in South Asia, three species in West Asia, two species in Central Asia, and two species in North America [3,7,48–52].’ I count 88. I guess that this is a review of all subg. Cerasus species have been mentioned in literature. So, you should remove this sentence from the MM section and add it to the discussion section.

Line 405. They mention ‘collected from the field or obtained from DNA Bank of China’

Please, provide details about how many were collected from the field and how many were from the DNA Bank of China.

Lines 22 and 24. Replase Prinus with Prunus

Line 39. Replace biogeoraphy with biogeography 

Author Response

Point 1: Lines 221-223. Please provide a relative reference.

Response1: Thanks. I have provided a relative reference.

Point 2: Lines 227-229. Please provide a relative reference.

Response2: Thanks. I have provided a relative reference.

Point 3: Line 392: The authors mention ‘It seems premature to make formal taxonomic revision of subg. Cerasus either at sectional or at species ranks before a reliable phylogeny based on nuclear genomes (or genes) was available.’ I agree with this sentence. So, you should close the section with a perspective about your future work (genome phylogenetic analysis).

Response3: Thanks for your suggestion. I have depicted the future work at the end of this section as a perspective.

Point 4: Line 400. The authors mention ‘The 87 ingroup represented 55 species of subg. Cerasus, nine species from subg. Prunus, and three species from subg. Padus.’ Total 55+9+3= 67 form 87, 20 is missing, please incorporate the 20 that are missing. For example, 55 species of subg. Cerasus (75 cp genomes).

Response4: Thanks for your suggestion. I have revised it.

Point 5: Line 402. The authors mention ‘For subg. Cerasus species, there were 45 species distributed in mainland China, five species in island Taiwan, 11 species in Japan, six species in the Korean Peninsula, six species in Russia, four species in Europe, four species in South Asia, three species in West Asia, two species in Central Asia, and two species in North America [3,7,48–52].’ I count 88. I guess that this is a review of all subg. Cerasus species have been mentioned in literature. So, you should remove this sentence from the MM section and add it to the discussion section.

Response5: Thanks for your suggestion. I have removed the sentence to the Line 242.

Point 6: Line 405. They mention ‘collected from the field or obtained from DNA Bank of China’. Please, provide details about how many were collected from the field and how many were from the DNA Bank of China.

Response6: Thanks. I have write the sample numbers from the field and the DNA Bank of China behind them.

Point 7: Lines 22 and 24. Replase Prinus with Prunus.

Response7: Thanks. I have revised them.

Point 8: Line 39. Replace biogeoraphy with biogeography

Response8: Thanks. I have revised it.

Round 2

Reviewer 1 Report

Unfortunately, I can't agree with some statements:

Point 1: I would replace Figure 1 (which didn’t carry any important or new facts) with a figure representing SNP analysis along the plastome. The several data could be visualized, beside SNP (from table S4, S5) - Pi diversity (from Figure S3), diagnostic nucleotides and output from selection analysis.

Response1: Thanks for your suggestion. I have added sequence and variation information to Figure 1, including gene distribution, exon and intron, SNP and InDel density, Pi diversity, GC content, and chloroplast genome structure.

It was not corrected, the revised verison contains the same figure 1 as the first one.

Point 2: Table S1 is missing GenBank Ids for newly sequenced plastomes.

Response2: Thanks for your suggestion. I have submitted the SRA of newly sequenced plastomes to NCBI under the bioproject PRJNA1017114, but these data are processing and have not been released.

Once again, the newly sequence plastomes have to be deposited and released in GenBank, each separately. Of course additionaly authors could make SRA submission, but without released accesion numbers of plastome sequences (not a Bioproject) reviewers can't validated their findings.

Point 5: It would be nice to see some superbarcoding approach in this paper, because there is no word about the potential of the plastome in molecular delimitation of Prunus.

Response5: Thanks for your suggestion. For the superbarcoding approach in molecular delimitation of Prunus, we have another paper submitted to study and discuss, so we do not refer to it in this paper.

Well, IJMS is a high IF journal and publishing only simple variation analyses without additional analyses reduce possible audience of this works.

Author Response

Point 1: I would replace Figure 1 (which didn’t carry any important or new facts) with a figure representing SNP analysis along the plastome. The several data could be visualized, beside SNP (from table S4, S5) - Pi diversity (from Figure S3), diagnostic nucleotides and output from selection analysis.

Response1: Thanks for your suggestion. I have added sequence and variation information to Figure 1, including gene distribution, exon and intron, SNP and InDel density, Pi diversity, GC content, and chloroplast genome structure.

It was not corrected, the revised verison contains the same figure 1 as the first one.

Response1: Yes, I have replaced Figure 1 in the revision.

Point 2: Table S1 is missing GenBank Ids for newly sequenced plastomes.

Response2: Thanks for your suggestion. I have submitted the SRA of newly sequenced plastomes to NCBI under the bioproject PRJNA1017114, but these data are processing and have not been released.

Once again, the newly sequence plastomes have to be deposited and released in GenBank, each separately. Of course additionaly authors could make SRA submission, but without released accesion numbers of plastome sequences (not a Bioproject) reviewers can't validated their findings.

Response2: The newly sequence plastomes have been deposited and released in GenBank. The accession numbers can be found in Table S1.

Point 5: It would be nice to see some superbarcoding approach in this paper, because there is no word about the potential of the plastome in molecular delimitation of Prunus.

Response5: Thanks for your suggestion. For the superbarcoding approach in molecular delimitation of Prunus, we have another paper submitted to study and discuss, so we do not refer to it in this paper.

Well, IJMS is a high IF journal and publishing only simple variation analyses without additional analyses reduce possible audience of this works.

Response3: Thanks for your suggestion. I have written the  superbarcoding analysis in the line 169-177.

Round 3

Reviewer 1 Report

The revised version is slightly improved, especially in the data reporting part.

However plastid superbarcoding part is not developed, again. In the lines 169-177 authors described only basic variation parameters and identify variation hotspots, which is not equivalent to the diagnostic nucleotides. I would encourage to apply some barcoding statistic (ASAP partitions, diagnostic nucleotides, GMYC analysis) and point out optimal regions for species identification based on these statistics.

Round 4

Reviewer 1 Report

Accepted